# Cryo-EM structure of the varicella-zoster virus A-capsid

Junqing Sun[1,2,9], Congcong Liu [3,9], Ruchao Peng [2,9], Fu-Kun Zhang[4,9], Zhou Tong[1,2,9], Sheng Liu[2], Yi Shi [2,5], Zhennan Zhao[2,5], Wen-Bo Zeng [6], George Fu Gao [2,5], Hong-Jie Shen[4], Xiaoming Yang [7✉], Minhua Luo [5,6✉], Jianxun Qi[2,5✉] & Peiyi Wang[3,8✉]

Varicella-zoster virus (VZV), a member of the *Alphaherpesvirinae* subfamily, causes severe diseases in humans of all ages. The viral capsids play critical roles in herpesvirus infection, making them potential antiviral targets. Here, we present the 3.7-Å-resolution structure of the VZV A-capsid and define the molecular determinants underpinning the assembly of this complicated viral machinery. Overall, the VZV capsid has a similar architecture to that of other known herpesviruses. The major capsid protein (MCP) assembles into pentons and hexons, forming extensive intra- and inter-capsomer interaction networks that are further secured by the small capsid protein (SCP) and the heterotriplex. The structure reveals a pocket beneath the floor of MCP that could potentially be targeted by antiviral inhibitors. In addition, we identified two alphaherpesvirus-specific structural features in SCP and Tri1 proteins. These observations highlight the divergence of different herpesviruses and provide an important basis for developing antiviral drugs.

[1] Shanxi Academy of Advanced Research and Innovation, Taiyuan 030032, China. [2] CAS Key Laboratory of Pathogenic Microbiology and Immunology, Institute of Microbiology, Chinese Academy of Sciences (CAS), Beijing 100101, China. [3] Cryo-EM Centre, Southern University of Science and Technology, Shenzhen 515055, China. [4] Changchun Keygen Biological Products Co. Ltd., Changchun 130000, China. [5] University of Chinese Academy of Sciences, Beijing 100049, China. [6] State Key Laboratory of Virology, CAS Center for Excellence in Brain Science and Intelligence Technology, Center for Biosafety Mega-Science, Wuhan Institute of Virology, Chinese Academy of Sciences, Wuhan 430071, China. [7] China National Pharmaceutical Group Corporation, Beijing 100029, China. [8] Department of Biology, Southern University of Science and Technology, Shenzhen 515055, China. [9] These authors contributed equally: Junqing Sun, Congcong Liu, Ruchao Peng, Fu-Kun Zhang, Zhou Tong. ✉email: yangxiaoming@sinopharm.com; luomh@wh.iov.cn; jxqi@im.ac.cn; wangpy@sustech.edu.cn

Varicella-zoster virus (VZV) can establish lifelong persistent infections and cause severe diseases in humans of all ages[1–3]. The primary infection of VZV causes varicella or chickenpox in young children, where the symptoms in most patients usually cease within one or two weeks as a result of host immunity. However, in some patients with uncommon severe infections, the complications could result in inflammation of the visceral organs, including hepatitis, pneumonitis, and encephalitis, which could be life-threatening, especially in immunocompromised individuals[1,4,5]. Despite that most global populations are vaccinated in childhood, ~1.5 million hospitalizations and 30,000 deaths due to VZV-related diseases are estimated to occur worldwide annually[6].

VZV belongs to the *Alphaherpesvirinae* subfamily of *Herpesviridae*, a group of enveloped double-stranded DNA (dsDNA) viruses[7]. The highly organized dsDNA genome is enclosed in an icosahedral capsid shell comprised by multiple viral proteins, which protects the viral genome from being exposed to the extracellular environment or detected by immune sensors in the cytosol[8,9]. After virus entry, the genome-containing viral capsid is transported into the nucleus by molecular motors along the microtubule system, whereupon the genomic DNA is released for replication in the nuclear viral factory. The newly synthesized viral genome is packed into the nascent capsid shell, which is subsequently exported from the inner lamella of the nuclear membrane by egress. After the maturation process, the VZV progeny virion bud at the plasma membrane but remain associated at the cell surface, a phenomenon that is quite unique among all herpesviruses[7]. Because the viral fusogenic glycoproteins are expressed on both the viral envelope and the membrane of infected cells, cell–cell fusion is easily mediated to facilitate virus transmission via syncytia formation[7].

In the replication process, the viral capsid plays a central role in maintaining viral genome integrity and enabling virion assembly in a highly regulated manner[10]. At least three kinds of intermediate capsid particles (A-, B-, and C-capsids), corresponding to different capsid assembly states, have been isolated from herpesvirus-infected cells[10,11]. A-capsids are empty protein shells resulting from abortive DNA packaging. B-capsids consist of a proteinaceous core made up of scaffold proteins that support capsid assembly at the early stage[12]. It is postulated that the scaffold proteins are released after proteolysis maturation, which enables packing in of the DNA genome to produce the full capsid particle (C-capsid)[13,14]. Apart from protecting the genomic DNA, the viral capsid also facilitates its latent infection in neurons, where it is transported retrogradely along the axon from the nerve termini to cell bodies at the sensory ganglia to establish latency. Upon conditional reactivation, the capsid is, in turn, transported from the neuron cell bodies to the innervated dermatomes in an anterograde manner to cause herpes zoster (or shingles)[15,16]. The knowledge about the structure of viral capsids and their mechanisms of the assembly would provide an important basis for developing antiviral therapeutics. In recent years, a few high-resolution capsid structures of human-infecting herpesviruses have been reported, including the herpes simplex virus-1 (HSV-1), HSV-2, human cytomegalovirus (HCMV), human herpesvirus-6B (HHV-6B), and Kaposi's sarcoma herpesvirus (KSHV)[17–21]. These studies revealed that the herpesvirus capsids have a similar overall architecture but display some unique features as well[17–21], suggesting the structural diversity of the herpesvirus family.

In this work, we present the 3.7-Å-resolution structure of the VZV A-capsid determined by cryogenic electron microscopy (cryo-EM) single-particle reconstruction. This structure clearly resolved the four major capsid components and revealed how these proteins interact with one another to maintain the stable viral capsid assembly. These data enhance our understanding on the structural conservation and divergence of various herpesviruses and provide clues for antiviral intervention.

## Results

**Architecture of VZV capsid shell.** The VZV capsid is a large icosahedron with a diameter of approximately 125 nm, which presents a significant challenge for high-resolution structure determination. The purified VZV capsids consisted mainly of A-capsid particles, with a small proportion of B-capsids and a few genome-containing C-capsids (Supplementary Fig. 1a). The initial reconstruction using conventional icosahedral averaging resulted in a density map at 6 Å resolution (Supplementary Fig. 1). To counteract the effects of local defocus variation and conformational heterogeneity within the large capsid particle, localized reconstruction of the sub-particles was carried out to resolve better details within an asymmetric unit (ASU)[22]. This strategy improved the resolution of the ASU block to 3.7 Å, in which most regions of the capsid floor reached a local resolution of 3.5 Å and the apical regions of the capsomer spikes revealed a relatively lower resolution of 4.0–4.5 Å (Supplementary Figs. 1 and 2). The 5-fold vertex displayed the most significant conformational heterogeneity and was moderately resolved at a local resolution of approximately 5.5 Å. In spite of the limited resolution in certain areas, most of the core regions provided sufficient details of the bulky side chains, thus allowing the atomic modeling of each structural component in the capsid shell (Supplementary Figs. 2–5 and Supplementary Table 1).

Similar to other herpesvirus capsids, the VZV capsid is composed mainly of four types of protein that are arranged in icosahedral symmetry with a triangulation (T) number of 16, including the major capsid protein (MCP), the small capsid protein (SCP), and the Tri1 and Tri2 proteins that make up the heterotriplex (Fig. 1). Each ASU contains 16 MCPs, 15 SCPs, and 5 heterotriplexes (Ta–Te). The MCPs assemble into either pentamers (pentons) at the 5-fold vertex or hexamers (hexons) in each facet of the icosahedron. According to the different molecular environments, there are three types of quasi-equivalent hexons in the capsid; namely, peripentonal (P), edge (E), and center (C) hexons (Fig. 1b, c). The MCPs in the different hexons basically adopt similar conformations, whereas the penton MCPs display a distinct conformation as a result of different assembly patterns (Fig. 1d). The conformational adaptation of the MCPs allows the stable viral capsid assembly to neutralize its internal pressure for genome packaging.

Among the twelve 5-fold vertexes, eleven of them are occupied by pentons, whereas the remaining unique vertex is a portal complex that allows the DNA genome to be packed in for assembly or released for replication[23]. To resolve the structure of this special portal vertex, sub-particles of the 5-fold vertexes were extracted by signal subtraction and further classified to isolate the portal complex from other penton capsomers. Unfortunately, we could not obtain distinctive 3D classes for the portal vertex after intensive classification (Supplementary Figs. 5 and 6), which was likely due to the absence of the DNA genome in the A-capsid. In the structures of the HSV-1[24,25], KSHV[26], and Epstein-Barr virus (EBV)[27] portals, clear density for the terminal DNA fragment is observed in the central channel, which forms extensive interactions with the portal proteins and may thus stabilize the conformation of the portal complex. These observations suggest the conformational flexibility of the portal complex in the intermediate capsids before genome packaging.

**Structures of the penton and hexon spikes.** The viral MCP displays an elongated shape and comprises a basal floor and a

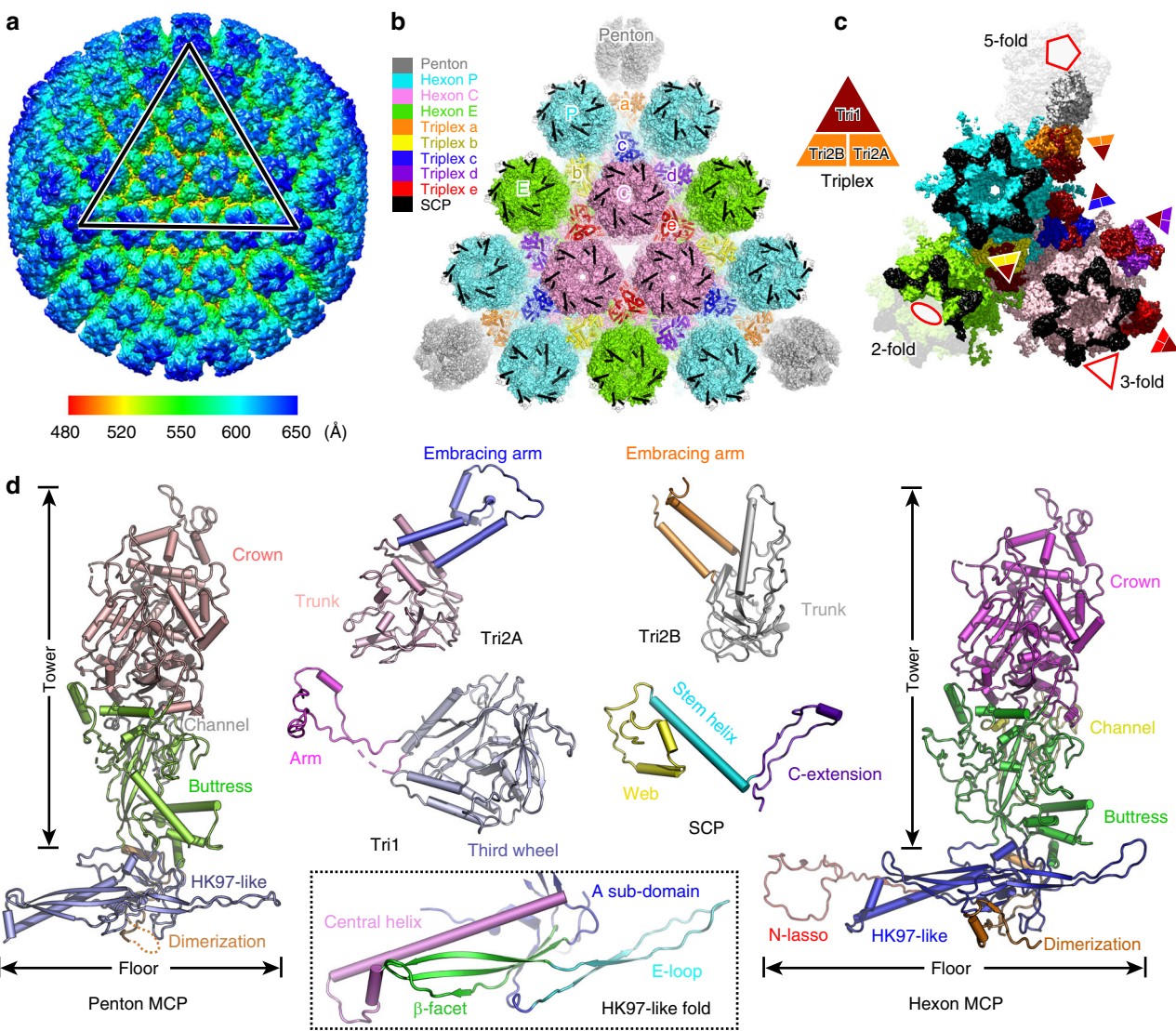

**Fig. 1 Overall architecture of VZV A-capsid. a** Density map of the icosahedral VZV A-capsid, colored by radius with the scheme below. An icosahedral facet is indicated by a triangle. **b** Organization of the capsid proteins within a facet. The symmetry-related molecules are represented by the same color. There are three types of hexons (P, C, and E) and five heterotriplexes (a–e) with unique molecular contexts in the capsid. The MCP is shown in surface, and the SCP and heterotriplex proteins are shown as cartoon models. **c** The density map of an ASU, colored by proteins. The MCPs in different capsomers are colored in the same scheme as used in (**b**). A schematic diagram of the heterotriplex assembly is shown to facilitate visualization of its orientation at different sites. **d** Atomic structure of each capsid protein, colored by domains. The inset shows the close-up view of the HK97-like fold in the floor of MCP.

vertical tower. The basal floor consists mainly of a HK97-like fold domain[28], which maintains the interaction networks of both intra- and inter-penton/-hexon capsomers (Fig. 1d). This domain was initially identified in the MCP (named gp5) of bacteriophage HK97[29] and has since been widely discovered in the capsid proteins of other bacteriophages[30,31] and herpesviruses[17–21,28]. Moreover, an N-terminal loop forms a lasso (N-lasso) at the floor layer to fasten the adjacent capsomers. The tower of the MCP is divided into two parts (i.e., the buttress neck domain and the upper crown), both of which participate in the assembly of individual penton and hexon capsomers (Figs. 1 and 2). On top of each hexon, six SCPs form a cap to cover the upper crown and to bridge adjacent MCP protomers for securing the hexameric structure. The stem helix of the SCP covers the crown apex of the corresponding MCP protomer, and its C-terminal extension protrudes into the crevice formed by the web domain and the crown of the adjacent SCP–MCP heterodimer (Fig. 2a, b). This interaction network helps to stabilize the contacting interface of

the upper crown of adjacent MCP protomers, thereby enhancing the stability of the hexons. By contrast, SCPs are not present in the penton capsomer of the VZV capsid. Instead, a large cliff is formed between two adjacent MCP protomers, resulting in few contacts between the buttress and crown domains (Fig. 2d, e). This observation partially explains the more severe conformational flexibility of pentons compared with hexons as revealed by the cryo-EM densities (Supplementary Fig. 2). In contrast to the SCPs of alphaherpesviruses, the SCPs of betaherpesviruses and gammaherpesviruses do bind to penton MCPs, but they do not crosslink the neighboring protomers[17,18,21,32]. Previous studies revealed that the absence of SCPs greatly impaired the assembly of the EBV[33] and KSHV[34] capsids in vitro and severely reduced the viral titer of HSV-1[35] and KSHV[32] in cells, suggesting the important role of the SCPs in herpesvirus assembly by stabilizing the hexons.

At the basal floor region, the HK97-like domains of adjacent MCP protomers intersect with one another to form a locked

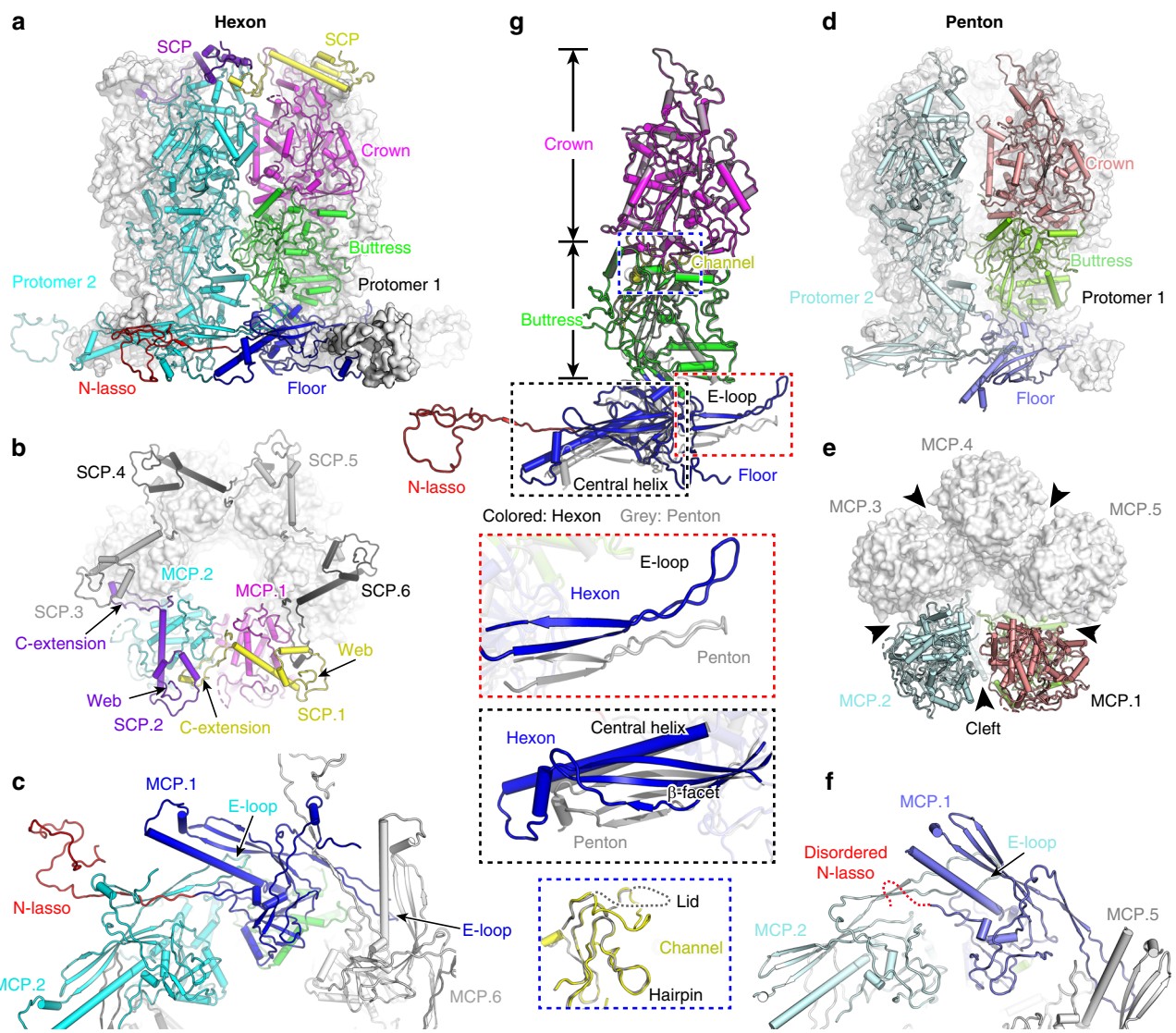

**Fig. 2 Structures of MCP and SCP in different capsomers. a** Overall structure of a hexon at the side view. Two adjacent MCP–SCP heterodimers are shown as cartoon models and the other four are shown as white surface models. One MCP is colored by domains and the other adjacent protomer is colored in cyan. The SCPs are colored by chains. **b** Top view of the hexon capsomer, revealing the contacting network of SCPs. **c** Bottom view of the basal floor interaction network in a hexon. **d–f** Architecture and inter-protomer interactions of a penton, shown in similar views and color schemes to those of the hexon in (**a–c**). **g** Superposition of the MCP conformers in hexon and penton capsomers. The insets show the close-up views of distinguished structural motifs with different conformations. The unresolved regions are represented by dashed lines.

circular network. The hairpin strands of the E-loop pair with the N-terminal strand of a second MCP to form a hybrid β-sheet, and the loop region interacts with the main β-facet of the adjacent MCP to further stabilize the basal network. Because of the different assemblies of pentons and hexons, the relative orientation of neighboring MCP protomers differs in the two types of capsomers, but they basically adopt similar contacting modes at the basal floor to maintain the stable assembly of each capsomer (Fig. 2c and f). Previous study reported a pyrazole derivative 35B2 as a specific VZV inhibitor that could impair viral capsid assembly in cells, and some escape mutations were observed in the N-terminal region of the MCP[36]. By mapping these sites onto the structure, we identified a hydrophobic pocket under the floor within the HK97-like domain (Supplementary Fig. 7). This pocket is not involved in either inter- or intra-capsomer interactions for capsid assembly, suggesting that 35B2 may interfere with the interactions between MCP and other binding partners to paralyze capsid formation in the early stage.

Comparing the MCP conformers in pentons and hexons, we found that the tower portion could be well superimposed, except for the channel domain, which revealed distinct conformations in the two capsomer forms (Fig. 2g). Two loop motifs in this domain are arranged in a circle within each capsomer, with the lower hairpin loop surrounding the channel walls and the lid loop covering the top (Supplementary Fig. 8). Owing to the distinct assembly patterns in pentons and hexons, the central channel in hexons displays a larger diameter than that in pentons. The lid loops in hexons intersect tightly with the neighboring molecules to strengthen the stability of the hexon spikes. By contrast, this region is disordered in the penton MCPs, resulting in an open roof for the penton channel (Supplementary Fig. 8). This conformational heterogeneity might allow the capsid to counteract the different pressures exerted on the icosahedral vertexes and facets by the packed DNA genome. The most remarkable difference between penton and hexon MCP conformers lies in the basal floor, which adapts to the different assembly networks

with 5-fold or 6-fold symmetry, respectively. Within the HK97-like domain, the central helix, β-facet, and E-loop of the penton MCPs all move downward relative to their counterparts in the hexons. Additionally, the N-lasso is well resolved in hexon MCPs, whereas it is disordered in the pentons, resulting in loss of the lasso interactions between the pentons and adjacent P-hexons (Fig. 2g). A similar disordered N-lasso is observed in the penton MCPs of the KSHV capsid[18]. For the HSV-1, HSV-2, HCMV, and HHV-6B capsids, however, the N-lasso of the penton MCP adopts a different conformation to that of the hexon MCP and contributes to the interactions between different capsomers around the penton[17,19–21].

**Interaction networks between capsomer spikes**. The assembly of penton and hexon capsomers is mediated mainly by the basal floor, which forms an enormous contacting network to stabilize the icosahedral cage of the capsid (Fig. 3). This interaction network involves a few different contacting modes at specific sites with distinct molecular contexts. Between adjacent hexons, the

basal floor forms symmetric contacting profiles involving four MCP protomers, as exemplified by the P- and C-hexons. The two strands in the E-loop of the C1 protomer are joined by a lower strand in the dimerization domain, an N-terminal strand of the C6 protomer, and one N-lasso strand of the P2 protomer to form a five-strand β-sheet that is further hooked up by the P2 N-lasso. The same contacts are found at the equivalent protomers related by C2 symmetry (Fig. 3b). Beneath the floor layer is the contacting network formed by the dimerization domain in which a helix and connecting loop are arranged into a rectangle with the opposite molecule. The dimerization domain could not be fully resolved, with approximately 30 residues missing in the density map, suggesting the flexibility of this region (Fig. 3c). Moreover, the N-lasso is also involved in interactions with the buttress domain above the basal floor. The hook region of the P2 N-lasso is accommodated by the chamber between the buttress domain and basal floor of the C1 protomer (Fig. 3d). These interactions highlight the central role of the N-lasso, which contributes to the extensive contacting networks with various domains of adjacent

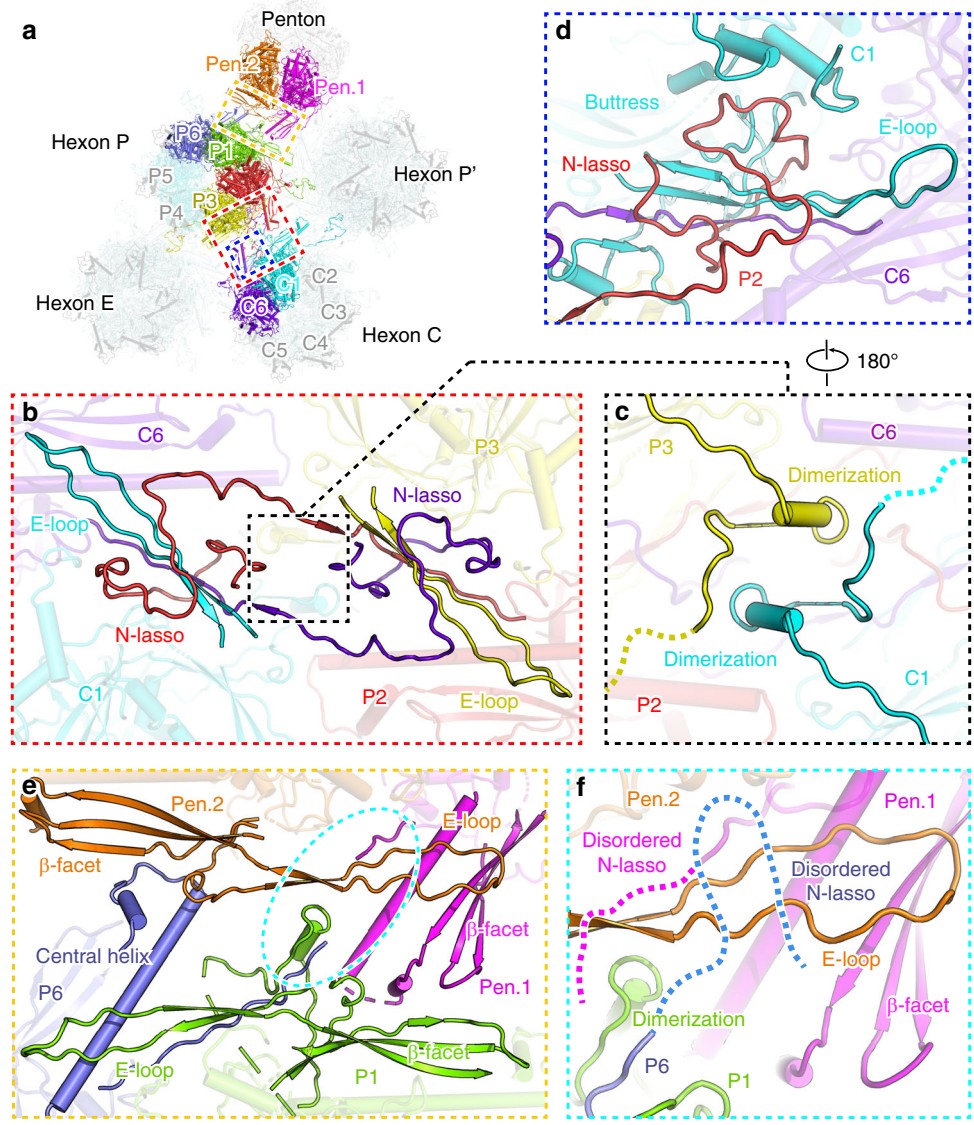

**Fig. 3 Inter-capsomer contacting networks at the basal floor. a** Overview of capsomer organization and interaction network within an ASU. The unique MCP conformers involved in inter-capsomer contacts are shown in different colors, and the rest parts are set transparent for clarity. **b–f** Close-up views of interactions between adjacent capsomers at different sites as indicated in (**a**). The critical motifs or domains involved in interactions are shown in solid colors, and the rest parts in the background are set transparent for clarity. The unresolved regions are represented by dashed lines.

MCPs. The deletion of this region in KSHV was found to be detrimental for virus viability, suggesting its essential role for capsid assembly in all herpesviruses[18].

With a different molecular context, the interaction patterns at the interface between penton and hexon capsomers are substantially different. As the relative orientation of two adjacent MCP protomers differs in hexons and pentons, the interactions between penton and hexon capsomers are not symmetric and involve multiple domains within the basal floor (Fig. 3e). The central helix and β-facet of the penton protomer Pen.1 protrude toward the β-facet strand of the P1 protomer in the center. The apical short helix and loop preceding the central long helix are in contact with the edge loop of the P1 β-facet. The P6 and Pen.2 protomers form a similar interaction profile by the side (Fig. 3e). Of note, the N-lasso of the penton and P6 MCPs are not observed in the density map, resulting in loss of lasso interactions between the pentons and P-hexons (Figs. 3f and 4). This phenomenon suggests that the N-lasso plays a more important role in hexon–hexon interactions than in penton–hexon contacts. Additionally, the dimerization domains of the penton MCPs and P1 protomer also display remarkable flexibility in the VZV capsid, in contrast to the stable dimerization interactions formed by two antiparallel helices as observed in the HSV-1 and HSV-2 capsids[19,20] (Figs. 3f and 4). These observations indicate that the penton–hexon interface of the VZV capsid is more flexible than that of the HSV-1, HSV-2, HHV-6B, and HCMV capsids, where substantive interactions are formed for the N-lasso and dimerization domains in these other viruses (Fig. 4). This flexibility might be compatible with the fact that VZV has the smallest genome of all human herpesviruses, which results in a relatively lower pressure being exerted onto the VZV capsid shell, in sharp contrast to the highly pressurized HCMV capsid with an almost 2-times larger genome than VZV (Supplementary Table 2).

**Structure of the heterotriplex**. Apart from the icosahedral cage formed by the basal floor, the inter-capsomer space is glued by additional heterotriplex proteins above the floor layer. Each heterotriplex is positioned at the interface of three capsomers and is surrounded by either three hexons or two hexons and a penton (Fig. 5a). The heterotriplex consists of one Tri1 and two Tri2-subunits that are assembled into a triangular-shaped structure. The two Tri2 molecules are organized in a face-to-face orientation and "embrace" each other via their arm helices. The embracing arms adopt different conformations in the two protomers, which are further crosslinked by an interchain disulfide bond. The groove at the dimeric interface is fastened by the stretching arm of the Tri1 molecule, resulting in an intersected heterotriplex (Fig. 5b–d). The two trunk domains of the Tri2 subunits and the third wheel of Tri1 form the three vertexes of the triangle, which insert into the space of two adjacent capsomers and interact with the basal floor and buttress domain of the MCP tower. For example, in the Tb heterotriplex, the Tri1 third wheel contacts mainly with the buttress domain of the MCP in the C-hexon; moreover, the trunk domain of Tri2A interacts with the floor and buttress domains of MCPs in both the E- and P-hexons, whereas the Tri2B subunit contributes the least contacts with the floor of the C-hexon (Fig. 5e–g). Of note, ~100 residues at the N-terminus (N-anchor) of the Tri1 subunit are not resolved in the density map (Fig. 5e), which are supposed to plug the holes in the floor and enhance the interactions between the heterotriplex and capsid layer. The Tri1 N-anchor is also highly flexible in the structure of the HSV-1 capsid[19] and is partially resolved in the HSV-2 capsid[20]. By contrast, it is well ordered in the HCMV, HHV-6B, and KSHV capsids[17,18,21]. The length of this motif is similar among the alphaherpesviruses and is longer than that in the betaherpesviruses and gammaherpesviruses. This difference possibly results in its lower rigidity in alphaherpesviruses, the capsids of which are less pressurized owing to the relatively smaller genomes compared with those of the other herpesviruses (Supplementary Table 2). The variation of the Tri1 N-anchor indicates its conformational adaptation to the different structural contexts in the capsomer networks and may also suggest a strategy adopted by different herpesviruses for counteracting the pressure exerted by the DNA genomes.

**Alphaherpesvirus-specific structural features of the capsid**. Generally, all herpesvirus capsids assemble in a similar architecture using conserved structural folds of the capsid proteins. However, there are also some specific local features that distinguish the different subfamilies of these viruses. Based on the available capsid structures of alphaherpesviruses (VZV, HSV-1, and HSV-2), betaherpesviruses (HCMV, HHV-6B), and one gammaherpesvirus (KSHV), we identified two major group-specific features for the alphaherpesviruses in the SCP and Tri1 proteins, respectively. Among the four capsid proteins, the MCP has the most conserved structures among all herpesviruses, which could be well superimposed with less than 2.5 Å root-mean-square deviations for the α-carbon atoms. By contrast, the SCP structures display quite obvious diversity, except for the shared stem helix in the center (Fig. 6a). For the alphaherpesvirus group, the N-terminal web of the SCP contains two small helices and a large portion of loop regions, whereas the same region in the betaherpesvirus and gammaherpesvirus SCPs is more orderly folded with longer helices. The most remarkable difference lies in the C-terminal structure, in which the alphaherpesvirus SCP contains a hairpin loop that crosslinks adjacent MCP–SCP heterodimers within each hexon capsomer. The C-terminus of the SCP of gammaherpesvirus KSHV forms a small bridging helix to contact with the adjacent SCP protomer, whereas the corresponding region in the betaherpesvirus SCPs is disordered and may thus contribute less to the interactions for stabilizing the hexons (Fig. 6a). Moreover, although SCPs are also present in the pentons of betaherpesviruses and gammaherpesviruses, they do not crosslink the neighboring MCPs[17,18,21]. The binding orientation of the central stem helix to the MCP crown is quite different between the alphaherpesviruses and the other two herpesvirus subfamilies. The helix in VZV, HSV-1, and HSV-2 SCPs is obviously tilted relative to the basal floor, whereas it binds with a smaller tilt angle in the structures of KSHV, HCMV, and HHV-6B, which might thus adapt to the different C-terminus structures for stabilizing the hexon capsomers in different viruses (Fig. 6a). Intriguingly, the VZV SCP harbors an unusually long C-terminal tail after the extension region, which could not be resolved in the density map. A similar flexible C-tail is also present in the SCP of the gammaherpesvirus, but it is much shorter in KSHV than in VZV (Supplementary Fig. 9a). A previous study has revealed that the extreme C-terminus of the VZV SCP contains a unique nuclear localization signal that is essential for the nuclear import of the MCP–SCP heterodimer[37]. Our structure suggests that this region is highly flexible and fully surface exposed to comply with the function of nuclear trafficking.

The structural folds of the heterotriplex proteins in different herpesviruses are relatively conserved, albeit with slight differences in the inter-subunit assembly contacts. In both HSV-1 and HSV-2, the two Tri2 molecules are non-covalently linked, whereas a disulfide bond is formed between Tri1 and Tri2B in the trunk and wheel domains[19,20]. By contrast, the two embracing arms of the Tri2A/Tri2B subunits in VZV, HCMV, HHV-6B, and

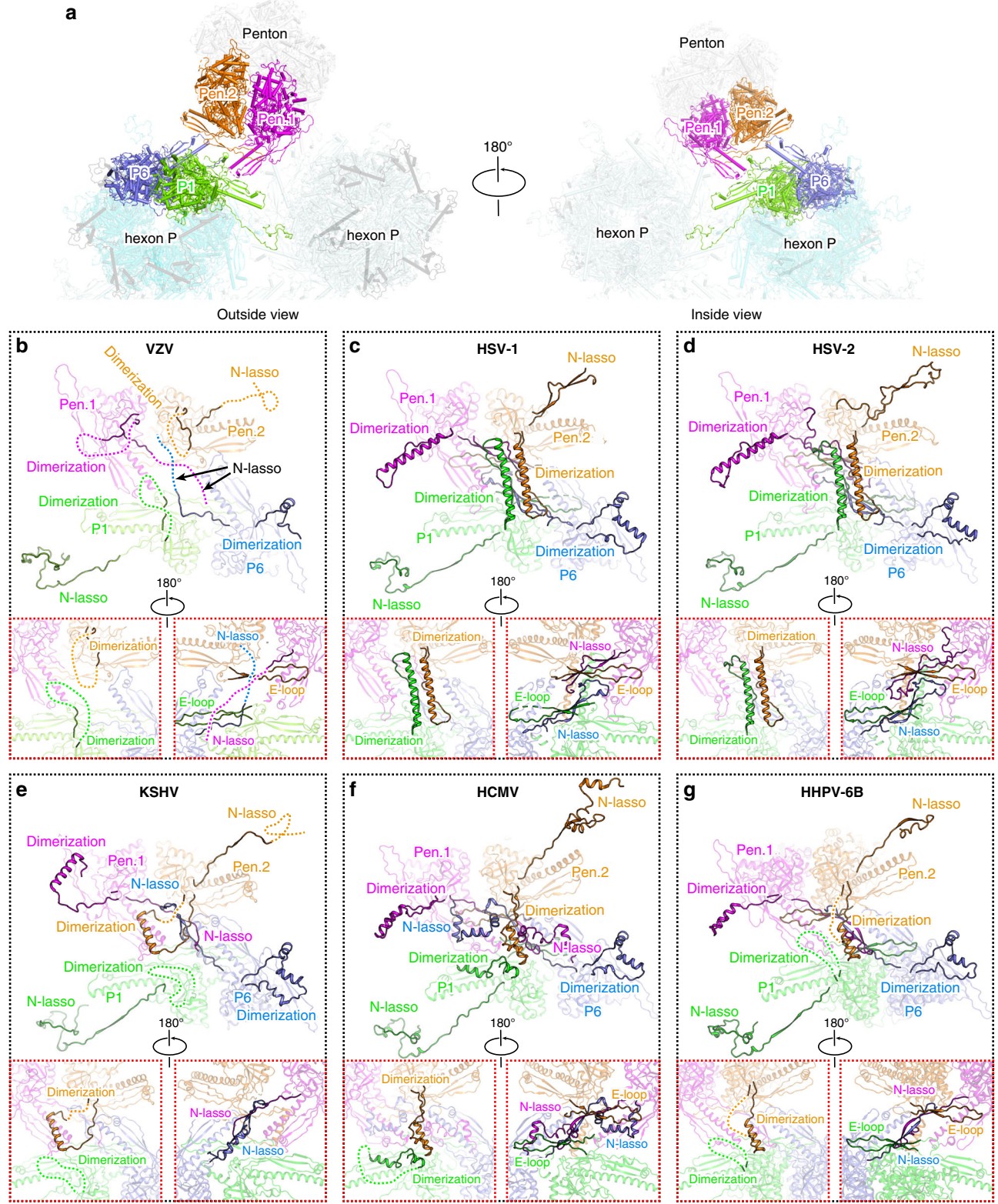

**Fig. 4 Comparison of the interaction networks between penton and hexon capsomers in different herpesviruses. a** Overview of the contacting interface between penton and hexon capsomers as viewed from outside or inside of the capsid. The four MCP conformers involved in interactions are highlighted in different colors. **b–g** Close-up views of the interactions between penton and hexon MCPs. The motifs involved in contacts are shown in solid colors, and the rest parts of the subunits are set transparent for clarity. The unresolved regions in the structure are represented by dashed lines. The insets show the zoomed-in inside (left panel) and outside (right panel) views of the central interaction details.

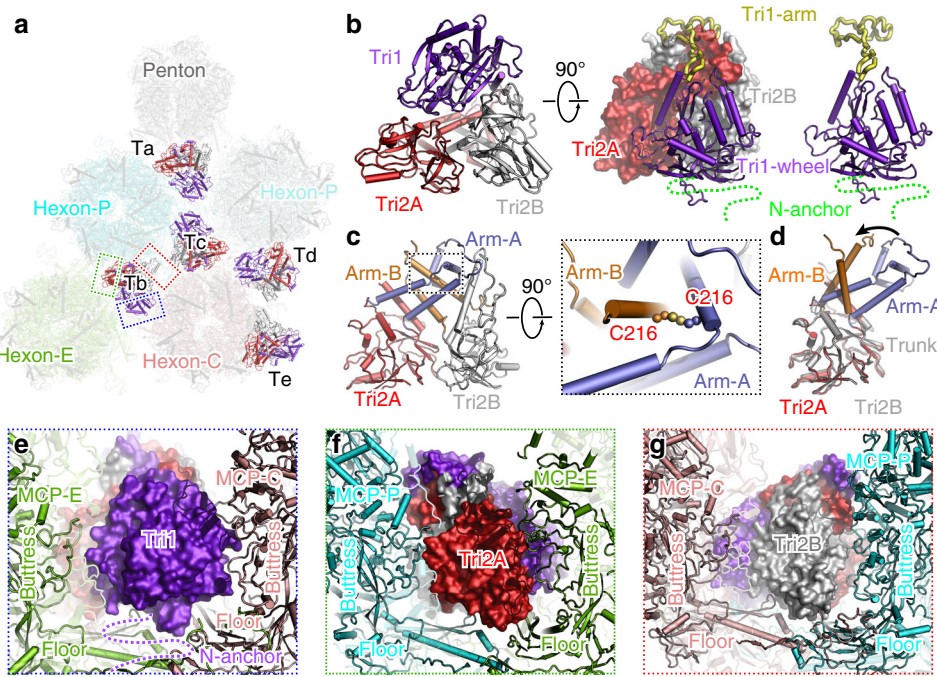

**Fig. 5 Structure of the glue heterotriplex. a** Overview of the heterotriplex distribution within an ASU in the capsid. The penton and hexon capsomers are set transparent to highlight the locations of the heterotriplex (colored by chains) at different sites. **b** Enlarged view of the Tri1–Tri2 heterotriplex structure, colored by chains. The internal insertion arm of Tri1 is highlighted in yellow and the disordered N-terminal anchor is represented with dashed lines. **c** Interactions between the two Tri2 subunits within the heterotriplex. The interchain disulfide bond between the embracing arms is shown as spheres in the enlarged box. **d** Comparison of the different conformations of the two Tri2 subunits in a heterotriplex. **e**–**g** Interactions between the heterotriplex (surface model) and capsomers (cartoon model).

KSHV are linked by one or two disulfide bonds without additional covalent interactions with the Tri1 molecule[17,18,21]. For the alphaherpesvirus capsid, the Tri1 subunit contains an internal insertion arm that binds to the groove between the embracing arms of Tri2A and Tri2B (Fig. 5b and Supplementary Fig. 9). However, the Tri1 molecules in the betaherpesviruses and gammaherpesvirus lack such a structural motif to assist the assembly of the heterotriplex. These observations highlight the diversity of the molecular determinants underpinning the capsid assembly of different herpesviruses, indicating that such group-specific features should be considered for the development of specific or broad-spectrum antivirals for the viruses in different subfamilies.

## Discussion

Given that the viral capsid plays a central role in herpesvirus infection and replication, elucidating the structural composition and assembly mechanism of the large capsid is essential for aiding the development of antivirals. The high-resolution structure of the VZV A-capsid contributes important knowledge toward a comprehensive understanding of herpesvirus replication and evolution. On the basis of the KSHV capsid structure, it has been established that a SCP-mimicking peptide could be engineered as a potent inhibitor of gammaherpesvirus replication[18]. The betaherpesvirus-specific tegument protein pp150 was also indicated as a potential immunogen for vaccine development[38–40]. Our systematic comparison of different herpesvirus capsid structures has revealed the conservation and divergence of potential drug targets, such as the SCP and Tri1 proteins, providing important information for the development of specific or broad-spectrum drugs against herpesviruses.

Similar to other herpesviruses, the scaffold proteins in VZV guide the nuclear import of newly synthesized MCP subunits[41].

However, in VZV, the SCP-dependent transportation is a non-redundant alternative means for the nuclear import of MCPs[37], which might be unique among all human herpesviruses. Therefore, the C-tail of SCP could be a promising target for developing specific antivirals to inhibit VZV capsid assembly by intercepting the transportation of viral proteins. It remains to be determined whether the insertion arm of Tri1 is indispensable for viral replication. Further functional studies at the virus level are required to test the feasibility of this strategy for drug targeting. A previously reported inhibitory compound, 35B2, impairs VZV capsid assembly in cells by targeting the MCP at the floor region[36]. Our structure suggests a pocket beneath the HK97-like fold as the potential binding site for this compound. Biochemical assays revealed that although 35B2 did not affect the nuclear import of MCPs from the cytoplasm, it did lead to their abnormal localization in the nucleus. This effect could be reverted through overexpression of the scaffold proteins[36], suggesting that the 35B2-binding site is crucial for the recognition between the individual MCP/scaffold subcomplexes to enable capsid assembly. This pocket may constitute a promising target for the development of highly potent VZV-specific antivirals.

An important feature of the herpesvirus capsid is its pseudo-icosahedral symmetry, resulting from the presence of the portal complex at a unique vertex. This complex governs the packaging and release of the DNA genome to facilitate virion assembly and replication[10]. In spite of great efforts, we were unable to reconstruct the portal complex within the VZV A-capsid. By reviewing the available portal structures of different herpesviruses, we realized that this complex might be quite flexible to serve as a switch with high sensitivity for DNA. In the context of the empty A-capsid, the conformational heterogeneity of the portal complex could be the reason to hinder its structural visualization. By contrast, the presence of the DNA fragment in the portal channel

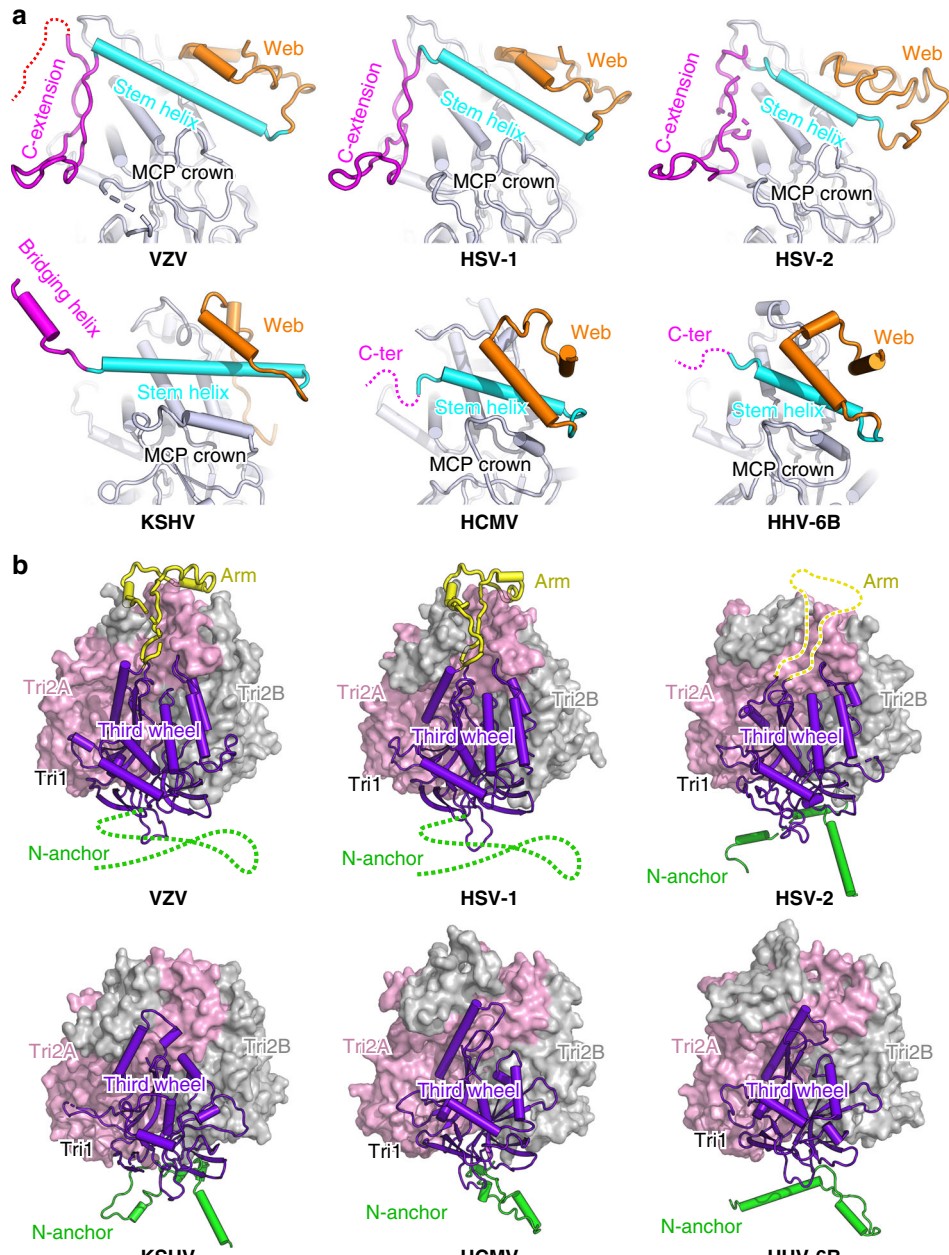

**Fig. 6 Group-specific structural features in herpesvirus capsid proteins. a** Comparison of SCP in different herpesviruses and its binding to the MCP. The MCP is colored in gray and the SCP is colored by domains. The disordered regions are represented by dashed lines. **b** Structures of the heterotriplex in different herpesviruses. The two Tri2 subunits are shown as surface models (colored by chains) and the Tri1 molecule is shown as a cartoon model (colored by domains). The Tri1 N-anchor in VZV and HSV-1 and the insertion arm in HSV-2, which are represented by dashed lines, were not resolved in the structure.

may stabilize its conformation, allowing the identification of this unique vertex from other penton vertexes. Further efforts will be made to resolve the portal structure in the C-capsid and to analyze the structural dynamics at different stages of VZV capsid assembly.

The herpesvirus capsid is decorated by a tegument layer, which enables its trafficking between subcellular compartments, nuclear egress, and establishment of latency[38,42]. The tegument proteins differ significantly among the various herpesviruses and are not yet fully defined for VZV. Our reconstruction did not reveal the density for the tegument components, possibly because of the different maturation states of the viral capsid in host cells. Previous studies have shown that the tegument complex exhibited

high occupancy in the C-capsids of HSV-1[43] and HSV-2[44], whereas it was almost invisible in the A- and B-capsids[20], suggesting the highly regulated process of viral capsid assembly. The low abundance of the VZV C-capsid in our preparation suggests that it has a long maturation process to facilitate its extraordinarily long latency in human cells.

In summary, we have determined the structure of the VZV A-capsid and analyzed the molecular determinants underpinning the assembly of this huge protein complex. Based on the available structures of related herpesvirus capsids, we have also identified the conserved and variable structural features of the different herpesvirus subfamilies. These findings indicate potential targets for antiviral drug design.

## Methods

**Cell culture and virus propagation and purification.** MRC-5 cells (ATCC CCL-171) were cultured to 100% confluency in Minimum Essential Medium supplemented with 10% newborn calf serum (Gibco). Then, the cells were infected with the VZV seed stock (strain vOka) at a multiplicity of infection of 0.01. At 3 days post infection, the cells were collected and resuspended in phosphate-buffered saline (1×PBS, pH 7.4) containing 2% NP-40 and lysed by three cycles of freezing and thawing. The lysate was centrifuged at $1500 \times g$ for 15 min to remove large debris. The supernatant was purified through a 15–60% (w/v) sucrose cushion, using an SW32 rotor for centrifugation at $80,000 \times g$ for 1.5 h. The virus-containing pellet was then resuspended in 1×PBS, loaded onto a linear 20–50% (w/v) sucrose density gradient, and centrifuged with an SW41 rotor at $80,000 \times g$ for 2 h. The band of virus particles was extracted using a syringe, diluted with 1×PBS, and centrifuged at $80,000 \times g$ for 1.5 h. The pelleted viruses were finally resuspended in 100 μL of 1×PBS and used for cryo-EM sample preparation. All steps of the purification procedure were performed at 4 °C.

**Cryo-EM sample preparation and data collection.** A 3-μL aliquot of purified VZV capsid was placed on a freshly glow-discharged lacey carbon grid. After 2.5 s of blotting with filter papers (Whatman #1), the grid was flash plunged in liquid ethane using a Vitrobot Mark IV system (Thermo Fisher Scientific Inc.) at 6 °C and 100% humidity. The cryo-EM data were collected using a Titan Krios microscope (Thermo Fisher Scientific Inc.) operated at 300 kV for automated image acquisition with EPU software. The images were recorded using a Falcon III detector, with a nominal magnification of ×59,000, resulting in a pixel size of 1.41 Å. Each exposure was performed with an accumulative dose of 40 e⁻/Å², which was fractionated into 39 frames for each image stack. The final defocus range of the micrographs was −0.8 to −1.5 μm.

**Image processing.** In total, 32,606 movie stacks of the VZV A-capsid were collected. The movie frames were aligned using MotionCor2[45] to correct the beam-induced drift. The contrast transfer function (CTF) parameters were estimated using CTFFIND4.1[46]. After excluding the micrographs with bad Thon rings, 26,979 micrographs were selected for subsequent processing. In total, 94,640 particles were boxed using the EMAN2 software package[47] and initially extracted as 4× binned particle images using RELION-3.0[48]. After several rounds of reference-free 2D classification, a subset of 72,247 particles was isolated for 3D classification and reconstruction. A map at 11.28 Å resolution was obtained through icosahedral averaging with 42,136 good particles. The coordinates were used to extract the 2× binned particles for 3D refinement using RELION-3.0 with I3 symmetry imposed, which resulted in a map at 6.67 Å resolution. The refined particles were extracted with the relion2lst.py script (https://github.com/homurachan/Block-based-reconstruction) to convert the data from a RELION project into the basic EMAN2 list format. The jalign program from JSPR-2017[49] was then used to further refine the Euler angle and center parameters, which enabled the reconstruction of a better map at 6.02 Å resolution using the j3dr program in JSPR-2017. To overcome the defocus gradient in this large capsid particle, block-based reconstruction was performed to further improve the resolution[22]. To this end, sub-particles for an asymmetric unit were clipped out and the new defocus and center parameters were calculated using the df_change_for_symmetric_unit_debug1.py and remove_misc_in_lst-center.py scripts from GitHub (https://github.com/homurachan/Block-based-reconstruction). The resultant sub-particles were imported into RELION-3.0 for 3D classification without applying symmetry, which led to a distinguished class with 320,882 sub-particles. The unbinned images were then extracted and subjected to 3D refinement using RELION-3.0, which generated a reconstruction at 4.7 Å resolution. At this stage, CTF refinement was carried out in RELION-3.0 to better estimate the local defocus values for each sub-particle. A final round of 3D refinement generated a density map at 3.9 Å resolution. A solvent mask was then applied to sharpen the map with an automatically estimated B-factor in RELION-3.0, resulting in a resolution of 3.7 Å as estimated by the gold-standard Fourier shell correlation (FSC) cut-off value of 0.143 (Supplementary Fig. 1). The local resolution distribution of the final reconstruction was assessed using ResMap[50] (Supplementary Fig. 2).

Because the VZV capsid contains a unique portal vertex, we extracted the sub-particles for the 5-fold vertexes to perform 3D classification in RELION-3.0 (Supplementary Fig. 5). This classification was conducted without rotation searching, with only offset refinement within a range of ±4 pixels. The density map of the vertex resulting from icosahedral reconstruction was clipped out and low-pass filtered to 30 Å to be used as the initial reference for 3D classification. The sub-particles were initially classified into four classes with C5 symmetry imposed. However, we did not identify any well-defined 3D classes for the portal vertex after extensive classification (Supplementary Fig. 5). We also attempted the classification of intact capsid particles using 4× binned images with C1 symmetry. Again, no portal-like density was observed at any vertexes (Supplementary Fig. 6). These observations suggest that the portal complex might be flexible in the empty A-capsid, which might be stabilized by dsDNA.

**Model building and refinement.** The structure of the HSV-2 capsid (PDB ID: 5ZZ8) was used as the initial model to fit in the density map of the VZV capsid with CHIMERA[51]. The model was manually adjusted in COOT[52] to improve the local fit and update the sequence register. The completed model was further improved through iterative positional and B-factor refinement using Phenix[53] and rebuilding in COOT. The stereochemical quality of the model was assessed using MolProbity[54]. The data collection, reconstruction and model refinement statistics are summarized in Supplementary Table 1. The structural figures were prepared using either PyMOL (http://www.pymol.org) or CHIMERA.

**Reporting summary.** Further information on research design is available in the Nature Research Reporting Summary linked to this article.

## Data availability

The data that support this work is available from the corresponding authors upon reasonable request. The cryo-EM density map of the VZV A-capsid and the atomic model of an ASU have been deposited to the Electron Microscopy Data Bank (EMDB) and Protein Data Bank (PDB) under the accession codes EMD-30228 and 7BW6, respectively.

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

## Acknowledgements

We thank all staff members of the Cryo-EM Centre, Southern University of Science and Technology for their assistance with data collection. This work was supported by the Strategic Priority Research Program of CAS (XDB37030204), National Natural Science Foundation of China (NSFC) (81673358), and the Technological Innovation Project of Shanxi Transformation and Comprehensive Reform Demonstration Zone (2017KJCX01). P.W. is supported by a startup grant from the Southern University of Science and Technology.

## Author contributions

P.W., J.Q., M.L., and X.Y. conceived and supervised the study. J.S., F.Z., Z.T., H.S., and W.Z. performed virus cultivation and purification. J.S., S.L., and Z.Z. prepared the cryo-EM grids and collected data. C.L. carried out data processing and reconstruction. R.P. and J.Q. conducted map segmentation and built the atomic model. C.L., R.P., Y.S., G.F.G., J.Q., and P.W. analyzed the structure. J.S., C.L., R.P., Y.S., and J.Q. prepared the figures and wrote the paper. All authors participated in discussions and paper editing.

## Competing interests

The authors declare no competing interests.
