## [Peer Review File · Nature Communications]

REVIEWERS' COMMENTS:

Reviewer #1 (Remarks to the Author):

Sun et al have submitted a revised version of their manuscript originally reviewed for NSMB that is now being considered by Nature Communications. I am reviewing it for a second time with the new journal in mind. I found the first version of the manuscript comprehensive and well-illustrated but lacking in the novelty expected for NSMB. The revision has addressed some of the concerns of the referees (provided primarily by referee #1) from the previous submission. In general, this is the best illustrated herpes virus structure that I have seen, and it is improved over the previous version in that regard. The novelty of this submission, in my opinion, is the clarity of the structure presentation compared to previous alpha herpes virus structures. I do not believe that the details differ significantly from previously published results for alpha herpes viruse structures. There is still some awkward English usage that needs to be improved by a native English speaking editor.

Response to referees:

Reviewer's comments:

Reviewer #1 (Remarks to the Author):

Sun et al have submitted a revised version of their manuscript originally reviewed for NSMB that is now being considered by Nature Communications. I am reviewing it for a second time with the new journal in mind. I found the first version of the manuscript comprehensive and well-illustrated but lacking in the novelty expected for NSMB. The revision has addressed some of the concerns of the referees (provided primarily by referee #1) from the previous submission. In general, this is the best illustrated herpes virus structure that I have seen, and it is improved over the previous version in that regard. The novelty of this submission, in my opinion, is the clarity of the structure presentation compared to previous alpha herpes virus structures. I do not believe that the details differ significantly from previously published results for alpha herpes virus structures. There is still some awkward English usage that needs to be improved by a native English-speaking editor.

Response: Thank you very much for the positive comment on our work and for the constructive suggestions to improve the manuscript. We have further revised the English usage by a commercial language editing service. We believe the current version would be clear enough and friendly for reading.